# SwiftMedSAM: An Ultra-Lightweight Prompt-based Universal Medical Image Segmentation Model for Highly Constrained Environments

Youngbin Kong[1,2][0009−0002−6248−4269], Kwangtai Kim[2][0009−0009−1141−5985],
Seoi Jeong[2][0009−0009−3047−2889], Kyu Eun Lee[3][0000−0002−2354−3599], and
Hyoun-Joong Kong[2,4,5,⋆][0000−0001−5456−4862]

[1] Interdisciplinary Program in Bioenglineering, Graduate School, Seoul National University, Seoul, Republic of Korea

[2] Department of Transdisciplinary Medicine, Seoul National University Hospital, Seoul, Republic of Korea

[3] Department of Surgery, Seoul National University Hospital and College of Medicine, Seoul, Republic of Korea

[4] Department of Medicine, Seoul National University College of Medicine, Seoul, Republic of Korea

[5] Innovative Medical Technology Research Institute, Seoul National University Hospital, Seoul, Republic of Korea

gongcop7@snu.ac.kr

**Abstract.** Medical image segmentation is a crucial step for accurate diagnosis and treatment planning, as it provides quantitative information about anatomical structures and pathological lesions in various clinical scenarios. However, the existing methodologies have limitations in terms of their generalizability and computational efficiency. In this study, we propose SwiftMedSAM, an ultra-lightweight prompt-based general model, to enable efficient medical image segmentation even in resource-constrained environments. Based on LiteMedSAM, we significantly reduced the model size and computational complexity through the hyperparameter optimization of the image encoder and mask decoder components. The developed model shows remarkable segmentation performance across various imaging modalities and anatomical structures while enabling real-time inference in resource-limited computing environments. The experimental results demonstrate that SwiftMedSAM outperforms the existing methodologies in terms of the trade-off between accuracy and efficiency. SwiftMedSAM achieved a validation score of 0.75 on the validation dataset. Owing to its unprecedented generalizability and low computational cost, SwiftMedSAM is expected to enable high-quality medical image analysis in resource-constrained settings, thereby contributing to advancements in precision medicine and telemedicine.

**Keywords:** Medical Imaging Segmentation · Segment Anything · Deep Learning

---

⋆ corresponding author

## 1   Introduction

Medical image segmentation is a critical step in accurate diagnosis and treatment planning. It provides quantitative information about anatomical structures and pathological lesions in various clinical scenarios such as computer-aided diagnosis, surgical guidance, treatment monitoring, and patient follow-up. For example, accurate identification of the location, size, and boundaries of a tumor is essential for determining cancer staging, surgical planning, and radiation therapy. However, such segmentation tasks are highly complex and time-consuming, necessitating the development of automated high-performance segmentation models [34].

Driven by advancements in deep learning techniques, innovative achievements have been made in the field of medical image analysis in recent years, with significant progress in segmentation problems. Initially, transfer-learned CNN-based models such as U-Net [49] and V-Net [43] were predominant, and subsequently, the introduction of cutting-edge models such as Vision Transformer [15] and Swin Transformer [36] led to substantial accuracy improvements. However, most existing studies have been limited by a lack of generalizability as they employ architectures and training methods tailored to specific clinical tasks or datasets [17].

Active research has been conducted in the field of segmentation foundation models for prompt-based universal image segmentation. A representative model, the Segment Anything Model (SAM), has demonstrated the ability to effectively perform various general image segmentation tasks using a single model through prompt engineering. However, SAM is a heavy model, making it impractical for use in resource-constrained environments or edge devices. To address this issue, lightweight models such as MobileSAM [61] and EfficientViT-SAM [62] have been proposed; however, they are specialized for natural image datasets rather than medical images, which presents a limitation.

In response, MedSAM was introduced, fine-tuning the existing SAM model on an unprecedented large-scale dataset comprising over one million medical image-mask pairs, achieving remarkable performance in medical image segmentation. MedSAM underwent comprehensive experimental evaluation on 86 internal and 60 external validation tasks, encompassing various anatomical structures, pathological conditions, and medical imaging modalities. The results showed that MedSAM consistently outperformed the previous SOTA segmentation model, SAM, and exhibited performance on par with or superior to specialized models [25] trained on the same imaging modality.

However, MedSAM, with 93M parameters, is an extremely large model that requires significant computational resources, making it difficult to utilize in resource-constrained computing environments. To address this limitation, LiteMedSAM, a lightweight version of the original MedSAM, was proposed. It was trained in two stages: distilling a lightweight encoder from MedSAM's large image encoder and then fine-tuning the entire pipeline with the distilled encoder. Through this process, LiteMedSAM achieved a significant reduction in model size and computational complexity compared to MedSAM, enabling faster inference in resource-constrained settings.

CVPR 2024: SEGMENT ANYTHING IN MEDICAL IMAGES ON LAPTOP Challenge focuses on developing a prompt-based general model for medical image segmentation. This challenge provides a large-scale dataset comprising over 1,000,000 image-mask pairs, including 11 medical imaging modalities and more than 20 types of cancer. The goal is to develop a prompt-based universal segmentation model that can handle various medical image segmentation tasks while being computationally lightweight enough to run on edge devices such as laptops. In this study, we used LiteMedSAM as the baseline model and optimized the hyperparameters of the image encoder and mask decoder components to develop a more lightweight SwiftMedSAM. While leveraging the large-scale dataset provided, we further reduced the model size and computational complexity to enable real-time inference, even in resource-constrained computing environments.

The developed SwiftMedSAM model is expected to have a significantly reduced model size and computational cost compared to LiteMedSAM, enabling real-time inference in even more constrained environments. Through this research, we aim to further mitigate the generalizability-efficiency trade-off of existing methods and achieve high-quality medical image segmentation under highly limited computing resources.

## 2    Method

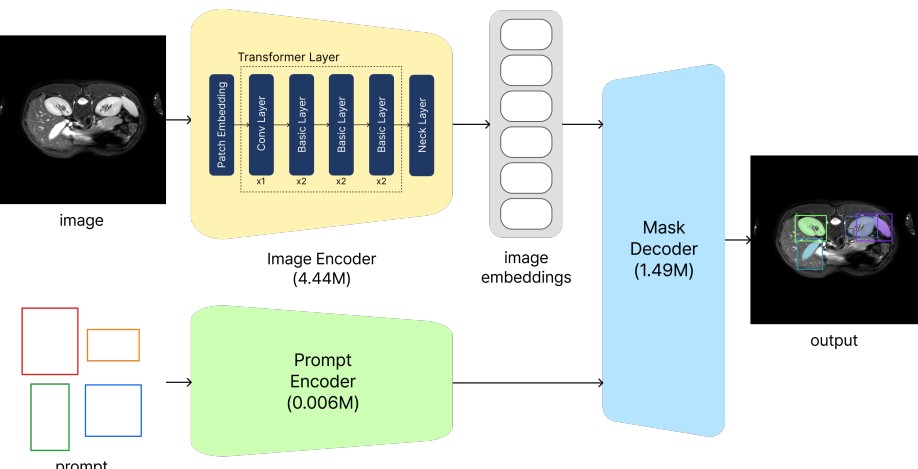

**Fig. 1.** Overall framework of the proposed method. The image and bounding box prompts serve as inputs to the model, passing through their respective encoders. The resulting outputs are then passed through the mask decoder to produce the segmentation results.

## 2.1    Preprocessing

The preprocessing strategy was inspired by MedSAM. We utilized a large-scale dataset with over one million image-mask pairs based on publicly available datasets that were used for MedSAM training. This dataset includes 11 imaging modalities (CT, MRI, endoscopy, ultrasound, etc.) and more than 30 types of cancers. The original 3D CT and MRI data, as well as grayscale images (X-ray, ultrasound, etc.) and RGB images (endoscopy, fundus, etc.), were converted to npz format for use. To structure the dataset and enable efficient management, the ground-truth masks and additional information, such as spacing, for both 2D and 3D images were stored together in a single file.

## 2.2    Proposed Method

The proposed SwiftMedSAM model builds upon the architecture of LiteMedSAM, with a focus on hyperparameter optimization and structural modifications to achieve ultra-lightweight performance. Our approach targets both the image encoder and mask decoder components, aiming to reduce computational load while maintaining high segmentation accuracy. The backbone of the existing LiteMedSAM was maintained, with the primary focus on hyperparameter optimization to achieve a balance between model efficiency and performance.
In the image encoder component, we made adjustments to the block depths to reduce the computational load. As the block depth increases, both the model capacity and computational complexity increase. Therefore, we employed a strategy of gradually decreasing the depths. In SwiftMedSAMv1, we applied block depths of [2, 2, 4, 2], whereas in SwiftMedSAMv2, we further reduced this to [1, 2, 2, 2]. This adjustment resulted in a lighter model while still preserving essential feature extraction capabilities.
For the mask decoder, we implemented several key modifications, primarily in the transformer and IoU head components. First, we reduced the transformer depth from 2 to 1, significantly decreasing the computational cost. Additionally, we made substantial reductions to the transformer's MLP dimensions, lowering them from the original 2048 to 1024, and further to 256. Larger MLP dimensions increase both model capacity and computational load, so reducing them contributes significantly to the lightening effect we aimed to achieve.
We also decreased the number of multi-head attention units in the transformer, reducing them from 8 to 4. While a higher number of attention heads allows for feature extraction from diverse perspectives, an excessive number risks overfitting and increases computational complexity. Therefore, we made an appropriate reduction to optimize both model capacity and computational load.
Lastly, we lowered the depth of the IoU head from 3 to 2, further reducing computations in this component. Although the IoU head, which is related to mask prediction, does not constitute a large portion of the overall computational load, we considered this adjustment valuable for our lightening purposes.
As a result of these optimizations, SwiftMedSAM has approximately 5.9M parameters, which represents a reduction of about 40% compared to the original

9.8M parameters of LiteMedSAM. This significant reduction in model size contributes to improved inference speed and reduced memory requirements, making SwiftMedSAM more suitable for deployment in resource-constrained environments.

To address the challenge of dataset imbalance, particularly the substantially lower number of PET modality images in the provided training dataset, we incorporated an additional dataset known as autoPETIII. This supplementary data was used to construct the final training dataset, helping to improve the model's performance across various imaging modalities, especially for PET images. The inclusion of this additional data helped to mitigate potential biases and enhance the model's generalization capabilities.

Through the combination of these structural lightening strategies and careful dataset curation, SwiftMedSAM achieves real-time performance and efficiency while maintaining high segmentation accuracy. The decrease in segmentation accuracy compared to the original model was not substantial, thanks to the pre-training on large-scale medical image data. This approach allows SwiftMedSAM to offer a compelling solution for medical image segmentation tasks in scenarios where computational resources are limited, without significantly compromising on the quality of the segmentation results.

**Table 1.** SwiftMedSAM Hyperparameters.

| Component | HyperParameters | Lite MedSAM | Swift MedSAMv1 | Swift MedSAMv2 |
|---|---|---|---|---|
| Image Encoder | Block Depths | [2, 2, 6, 2] | [2, 2, 4, 2] | [1, 2, 2, 2] |
| Mask Decoder | Transformer Depth | 2 | 1 | 1 |
| Mask Decoder | Transformer MLP Dim | 2048 | 1024 | 256 |
| Mask Decoder | Transformer Num Heads | 8 | 8 | 4 |
| Mask Decoder | IOU Head Depth | 3 | 2 | 2 |

### 2.3   Post-processing

The Swift MedSAM model proposed in this study includes a post-processing stage that converts the predicted mask to the original image size through a series of steps. This post-processing stage ensures that the mask output by the model is aligned with the original image size, thereby providing an accurate segmentation result. The post-processing stage consists of the following steps:

1. Cropping
   The predicted mask is resized to the size of the input image ($256 \times 256$) for the model, the mask undergoes a cropping process to eliminate unnecessary padding areas.

2. Resizing
   The cropped mask is resized to the original size of the image. This is achieved through bilinear interpolation, upsampling the mask to the same size as the original image.
3. Sigmoid Activation Function
   A sigmoid activation function is applied to the upsampled mask, normalizing the values of each pixel between 0 and 1. This step ensures that each pixel in the mask represents the probability of belonging to the target region.
4. Binarization
   The mask that undergoes the sigmoid activation function is binarized using a threshold of 0.5. In other words, values greater than or equal to 0.5 are converted to 1, and values below 0.5 are converted to 0, generating the final mask. This process ensures that the predicted mask has a clear binary form.

## 3    Experiments

### 3.1    Dataset and evaluation measures

In the CVPR 2024: SEGMENT ANYTHING IN MEDICAL IMAGES ON LAPTOP challenge, participants could use the training and validation datasets provided by the organizers and external publicly available datasets. The datasets used in developing SwiftMedSAM are as follows: COVID-19-20 [51], AbdomenCT-1K [40], FDG-PET-CT-Lesions [18], NSCLC Radiogenomics [6], NSCLC-Radiomics [18], CT Lymph Nodes [50], NSCLC-PleuralEffusion [31], NSCLC-Lung MSD-LUNG [53,4,38], KiTS23 [20], CT-ORG [4], COVID-19-20-CTSEG [38], TotalSegmentator [58], AMOS [28], LCTSC [60], HCC-TACE-Seg [45], Adrenal-ACC-Ki67-Seg [44], MSD [4,52], ISLES [21], WMH [33], BraTS [5,42], PROMISE12 [35], MSD-Prostate [4,52], NCI-ISBI [7], Cross-moda [14], QIN-PROSTATE-Repeatability [16], CC-Tumor Heterogeneity [8], COVID-19 Radiography Database [48,10], COVID-QU-Ex [55,13,10,48], Chest Xray Masks and Labels [9,26], Chest X-Ray Images with Pneumothorax Masks, CDD-CESM [30,29], Intraretinal Cystoid Fluid [2], ps-fh-aop-2023 [37], hc18 [22,23], Breast Ultrasound Images Dataset [3], ISIC2018 [56,11,12], Cholec-Seg8k [24,57], Kvasir-SEG [27,46], m2caiSeg [41], PAPILA [32], IDRiD [47], NeurIPS CellSeg [39], autoPETIII.

The training dataset includes 11 imaging modalities: Computed Tomography (CT), Magnetic Resonance Imaging (MRI), Positron Emission Tomography (PET), X-ray, ultrasound, mammography, Optical Coherence Tomography (OCT), endoscopy, fundus, dermoscopy, and microscopy. A total of 1,490,576 medical image-mask pairs were used to train our model. The validation dataset contains 9 modalities and is a subset of the testing set used in this challenge.

The evaluation metrics for this challenge were divided into accuracy and efficiency. The accuracy metrics are Dice Similarity Coefficient (DSC), which measures the overlap between the ground truth and predictions, and Normalized Surface Dice (NSD), which measures the similarity between the ground truth boundary and predictions. The efficiency metric is runtime, measured using only the CPU without GPU assistance.

### 3.2   Implementation details

**Environment settings**  The development environments and requirements are presented in Table 2.

**Table 2.** Development environments and requirements.

| | |
|---|---|
| System | Ubuntu 20.04.6 LTS |
| CPU | AMD EPYC™7402X CPU@2.8GHz |
| RAM | 8×64GB; 3200MT/s |
| GPU (number and type) | Four NVIDIA A100 80G |
| CUDA version | 11.8 |
| Programming language | Python 3.10.13 |
| Deep learning framework | torch 2.1.0 , torchvision 0.16.0 |
| Code | |

**Training protocols**  The training protocols of SwiftMedSAM is listed in Table 3.

**Table 3.** Training protocols.

| | |
|---|---|
| Pre-trained Model | LiteMedSAM |
| Batch size | 32 |
| Patch size | 256×256×3 |
| Total epochs | 26 |
| Optimizer | AdamW ($\beta_1 = 0.9$, $\beta_2 = 0.999$) |
| Initial learning rate (lr) | 0.005 |
| Lr decay schedule | ReduceLROnPlateau |
| Training time | 93.5 hours |
| Loss function | Dice Loss, Binary Cross Entrophy Loss |
| Number of model parameters | 5.94M[6] |
| Number of flops | 30.04G[7] |
| $CO_2$eq | 16.16 Kg[8] |

## 4   Results and discussion

### 4.1   Quantitative results on validation set

In this study, we conducted experiments to evaluate the performance of SwiftMed-SAM. The accuracy was measured based on the 3,076 validation data images provided and compared with the baseline model, LiteMedSAM. The results are presented in Table 4. Compared to the baseline, SwiftMedSAMv1 exhibited a 0.05% average decrease in DSC, but a 2.00% improvement in NSD. Compared to

SwiftMedSAMv1, SwiftMedSAMv2 showed a 0.63% improvement in DSC and 0.73% improvement in NSD.

In particular, when examining each imaging modality, for CT images, SwiftMedSAMv1 achieved a 2.38% improvement in DSC and a 3.39% improvement in NSD compared to the baseline. For MR images, DSC improved by 3.42% and NSD improved by 4.45%. For PET images, there was a significant improvement, with DSC increasing by 13.77% and NSD by 21.79%. However, for US images, DSC decreased by 15.61% and NSD decreased by 11.69%. For X-ray images, DSC decreased by 11.34% and NSD decreased by 9.55%.

Comparing SwiftMedSAMv2 and SwiftMedSAMv1, for CT images, DSC decreased by 0.33%, whereas NSD decreased by 0.27%. For MR images, DSC decreased by 0.78% and NSD decreased by 0.56%. For PET images, DSC decreased by 5.49% and NSD decreased by 0.93%. For US images, DSC improved by 1.72%, and NSD improved by 2.14%. For X-ray images, DSC improved by 6.24%, and NSD improved by 5.88%.

**Table 4.** Quantitative evaluation results on validation set.

| Target | Baseline | | Swift MedSAMv1 | | Swift MedSAMv2 | |
|---|---|---|---|---|---|---|
| | DSC(%) | NSD(%) | DSC(%) | NSD(%) | DSC(%) | NSD (%) |
| CT | 40.71 | 40.27 | 43.09 | 43.66 | 42.76 | 43.39 |
| MR | 61.17 | 62.40 | 64.59 | 66.85 | 63.81 | 66.29 |
| PET | 55.10 | 29.17 | 68.87 | 50.96 | 63.38 | 50.03 |
| US | 94.77 | 96.81 | 79.16 | 85.12 | 80.88 | 87.26 |
| X-Ray | 75.82 | 80.38 | 64.48 | 70.83 | 70.72 | 76.71 |
| Dermotology | 92.47 | 93.85 | 93.88 | 95.30 | 93.43 | 94.80 |
| Endoscopy | 96.04 | 98.11 | 94.57 | 97.22 | 95.58 | 98.02 |
| Fundus | 94.81 | 96.41 | 94.10 | 95.79 | 96.13 | 97.69 |
| Microscopy | 61.63 | 65.39 | 69.41 | 75.05 | 66.13 | 73.13 |
| Average | 74.73 | 73.64 | 74.68 | 75.64 | 75.31 | 76.37 |

### 4.2   Qualitative results on validation set

For the comparison of qualitative results, we used publicly available datasets with ground truth annotations: CT2USforKidneySeg [54], HipXRay [19], and NSCLC-Radiomics [1], which contain ultrasound, X-ray, and CT modalities, respectively. Examples of SwiftMedSAMv2's segmentation results for these datasets are shown in Fig 2. While SwiftMedSAMv2 demonstrated refined segmentation performance on the CT2USforKidneySeg and NSCLC-Radiomics datasets, it yielded suboptimal results for some images from the HipXRay dataset with low-contrast or unclear boundaries.

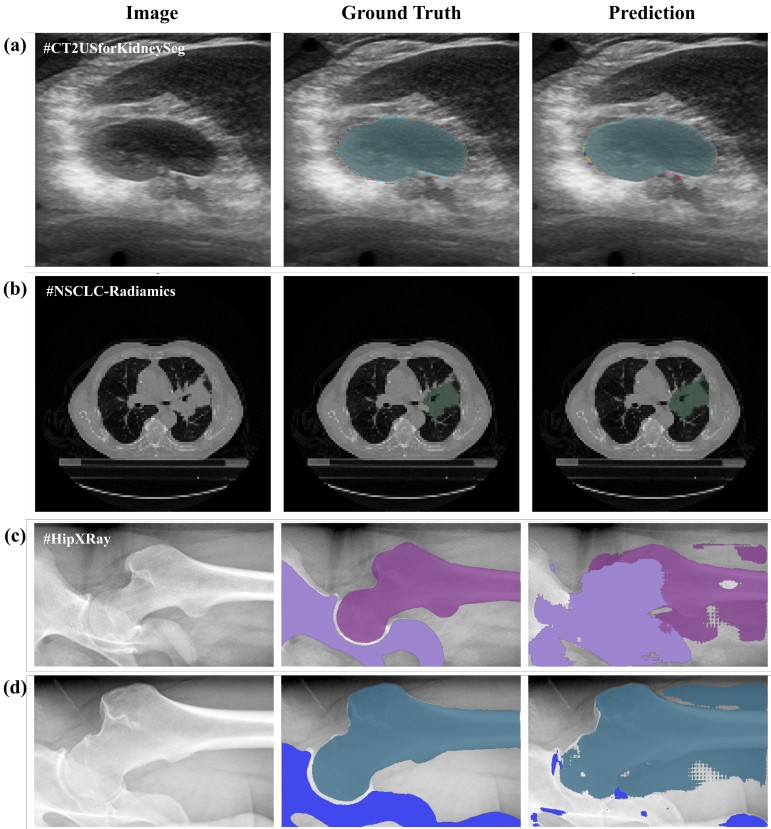

**Fig. 2.** Qualitative results of our SwiftMedSAMv2. (a-b) Good segmentation cases from CT2USforKidneySeg dataset and NSCLC-Radiomics dataset, respectively. Our method accurately delineates the kidney boundaries in the CT image (a) and captures the tumor region in the lung CT image (b). (c-d) Bad segmentation cases from the HipXRay dataset, where our method struggles to segment the femoral head and acetabulum regions precisely due to low contrast and complex anatomical structures.

### 4.3   Segmentation efficiency results on validation set

The efficiency experiments for the final model were performed on a CPU: AMD EPYC™7402X CPU@2.8GHz, RAM: 8×64GB; 3200MT/s. The specific inference time measurements for some cases are listed in Table 5. Consequently, the SwiftMedSAMv1 and SwiftMedSAMv2 models showed reduced execution times compared to the baseline model in most cases. Particularly, for the 3DBox_MR_0621 case, while the baseline model's execution time was 630.2 seconds, SwiftMedSAMv1 and SwiftMedSAMv2 took 166.8 seconds and 145.0 seconds, respectively, showing a significant reduction. Compared to SwiftMed-SAMv1, SwiftMedSAMv2 exhibited better performance in some cases, and an overall slight improvement was observed. For instance, in the 3DBox_CT_0566

case, SwiftMedSAMv1 recorded 343.2 seconds, while SwiftMedSAMv2 took 331.9 seconds, demonstrating a faster execution time.

**Table 5.** Quantitative evaluation of efficiency in terms of running time (s).

| Case ID | Size | Num. Objects | Baseline | Swift MedSAMv1 | Swift MedSAMv2 |
|---|---|---|---|---|---|
| 3DBox_CT_0566 | (287, 512, 512) | 6 | 499.2 | 343.2 | 331.9 |
| 3DBox_CT_0888 | (237, 512, 512) | 6 | 114.1 | 97.4 | 94.1 |
| 3DBox_CT_0860 | (246, 512, 512) | 1 | 17.7 | 15.9 | 14.2 |
| 3DBox_MR_0621 | (115, 400, 400) | 6 | 630.2 | 166.8 | 145.0 |
| 3DBox_MR_0121 | (64, 290, 320) | 6 | 119.4 | 94.7 | 90.7 |
| 3DBox_MR_0179 | (84, 512, 512) | 1 | 17.7 | 14.7 | 13.3 |
| 3DBox_PET_0001 | (264, 200, 200) | 1 | 31.9 | 9.9 | 8.6 |
| 2DBox_US_0525 | (256, 256, 3) | 1 | 1.8 | 1.7 | 1.6 |
| 2DBox_X-Ray_0053 | (320, 640, 3) | 34 | 9.8 | 9.3 | 9.8 |
| 2DBox_Dermoscopy_0003 | (3024, 4032, 3) | 1 | 7.9 | 8.4 | 7.9 |
| 2DBox_Endoscopy_0086 | (480, 560, 3) | 1 | 6.1 | 2.7 | 2.6 |
| 2DBox_Fundus_0003 | (2048, 2048, 3) | 1 | 2.6 | 4.2 | 4.0 |
| 2DBox_Microscope_0008 | (1536, 2040, 3) | 19 | 19.5 | 18.1 | 18.0 |
| 2DBox_Microscope_0016 | (1920, 2560, 3) | 241 | 257.5 | 253.1 | 267.1 |

### 4.4 Results on final testing set

The SwiftMedSAM was evaluated on the final testing set across various medical imaging modalities. In terms of segmentation accuracy metrics, endoscopy achieved the highest DSC of 91.55% and NSD of 94.44%. The average DSC and NSD across all modalities were 75.50% and 78.95%, respectively. Regarding efficiency, the mean runtime was 12.86 seconds, with endoscopy being the fastest at 7.37 seconds and CT the slowest at 30.89 seconds. Detailed results are presented in Table 6.

**Table 6.** Testing results on final testing set

| Target | SwiftMedSAM | | |
|---|---|---|---|
| | DSC(%) | NSD(%) | Runtime (s) |
| CT | 61.03 | 65.56 | 30.89 |
| MR | 66.73 | 68.62 | 14.51 |
| X-Ray | 64.55 | 77.04 | 9.25 |
| Endoscopy | 91.55 | 94.44 | 7.37 |
| Fundus | 85.96 | 88.15 | 8.96 |
| Microscopy | 81.00 | 83.00 | 15.73 |
| OCT | 71.09 | 78.47 | 7.84 |
| PET | 79.30 | 70.65 | 12.09 |
| US | 78.28 | 84.63 | 9.10 |
| Average | 75.50 | 78.95 | 12.86 |

### 4.5   Limitation and future work

SwiftMedSAM exhibited versatility in medical image segmentation despite its remarkably small model size and low computational complexity. However, there are still limitations in which segmentation errors occur when the boundaries of the structures/lesions are ambiguous. We expect that these issues can be resolved with higher-quality data and more powerful training strategies in the future. Currently, the performance is maintained at the level of the baseline model, but we aim to achieve a fast inference speed and improve the performance in the future.

## 5   Conclusion

In this study, we proposed SwiftMedSAM, an ultra-lightweight prompt-based model that enables real-time high-performance medical image segmentation even in highly constrained computing environments. While maintaining the backbone of the existing SOTA model MedSAM, we introduced the lightweight LiteMedSAM as the baseline and performed a process of hyperparameter tuning to drastically reduce the model size and computational complexity. Through experiments, we verified the comparable segmentation performance and fast inference speed of SwiftMedSAM.
The proposed SwiftMedSAM demonstrates the potential for a universal prompt-based medical image segmentation model while simultaneously pursuing efficiency and generalizability. This will enable high-quality medical image analysis, even in resource-constrained environments, contributing to advancements in precision medicine and telemedicine.

**Acknowledgements**  We thank all the data owners for making the medical images publicly available and CodaLab [59] for hosting the challenge platform.

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

**Table 7.** Checklist Table. Please fill out this checklist table in the answer column.

| Requirements | Answer |
| --- | --- |
| A meaningful title | Yes |
| The number of authors ($\leq$6) | 5 |
| Author affiliations and ORCID | Yes |
| Corresponding author email is presented | Yes |
| Validation scores are presented in the abstract | Yes |
| Introduction includes at least three parts: background, related work, and motivation | Yes |
| A pipeline/network figure is provided | Figure 1 |
| Pre-processing | Page 4 |
| Strategies to data augmentation | (none) |
| Strategies to improve model inference | Page 4 |
| Post-processing | Page 5 |
| Environment setting table is provided | Table 2 |
| Training protocol table is provided | Table 3 |
| Ablation study | Page 7 |
| Efficiency evaluation results are provided | Table 5 |
| Visualized segmentation example is provided | Figure 2 |
| Limitation and future work are presented | Yes |
| Reference format is consistent. | Yes |
| Main text $>=$ 8 pages (not include references and appendix) | Yes |