# OpenReview forum: "SwiftMedSAM: An Ultra-Lightweight Prompt-based Universal Medical Image Segmentation Model for Highly Constrained Environments"
_thecvf.com/CVPR/2024/Workshop/MedSAMonLaptop — CVPR24 MedSAMonLaptop_

### Official Review · Reviewer_oP3C · 2024-06-10
**Recommend**

**Rating:** 7
**Confidence:** 4

**Review:**

SwiftMedSAM presents an optimized model for medical image segmentation, designed to perform efficiently in resource-constrained settings. Building on LiteMedSAM, it significantly reduces model size and computational complexity through hyperparameter optimization of the image encoder and mask decoder components. Validation results indicate that SwiftMedSAM offers a trade-off between accuracy and efficiency, achieving a validation score of 0.75. The detailed methodology and extensive experimental results support the reproducibility and robustness of the model. I recommend this paper for acceptance due to its less computational resources with full details for reproducibility.

---

### Official Review · Reviewer_aGr9 · 2024-06-15
**SwiftMedSAM: An Ultra-Lightweight Prompt-based Universal Medical Image Segmentation Model for Highly Constrained Environments**

**Rating:** 4
**Confidence:** 4

**Review:**

Paper Summary:

The authors propose an approach to pruning the model size by performing a series of ablations starting from a LiteMedSAM baseline model. To achieve this goal, there are several levers to adjust: First, reduce the block depth of different blocks of the image encoder; second, they tackle the mask decoder by reducing the transformer depth from two layers to one layer and reducing its MLP dimensionality. On top, the authors reduce the number of attention heads from 8 to 4. Finally, the depth of the IoU-head was lowered from 3 to 2. All these changes are reflected in the architecture SwiftMedSAMv1. The authors also propose SwiftMedSAMv2, in which they prune even more aggressively.

Paper Strengths:

Pruning the model to an ideal size is a meaningful idea as it not only reduces training but also inference time.
Using additional allowed datasets (e.g., AutoPET III) to improve the balance of available images is a reasonable approach.

Paper Weaknesses:

The paper lacks key information in certain sections:
What do the authors exactly mean by “block depth”?
Which component are the authors referring to when talking about the IOU-head depth? Is this the MLP in the final stage that predicts the IoU scores?
Addressing these components via a figure would greatly facilitate the understanding of what the authors changed. Fig. 1 is not really helpful as it shows the architecture on too high a level.
The information on how the authors trained their pruned models is also rather short. Did they start with randomly initialized weights or use some form of image distillation? Did they use the weights of the pruned larger models?
Results: It’s a bit unclear how the results of the baseline as evaluated by the authors lead to such bad results. Assuming they used LiteMedSAM as provided by the organizers of the challenge, their reported results on the CT and MRI domains are drastically different from what others have reported (e.g., 40.71 for dice vs. ~92). In other domains e.g., Ultrasound, the results are similar. This raises the question of the evaluation protocol which the authors used and should be clarified.

Minor: spelling mistake in the first sentence of Sec. 2.2


Overall, I like the idea of pruning the model leading to a better tradeoff of accuracy and efficiency, but I feel the paper needs some improvements to better convey this idea and provide the necessary details to better understand what the authors did. Also, the large difference in the reported results is concerning.

---

### Official Review · Reviewer_yX3k · 2024-06-16
**Lacks supporting evidence and comparative analysis**

**Rating:** 3
**Confidence:** 4

**Review:**

The paper introduces SwiftMedSAM, a proposed ultra-lightweight model for medical image segmentation designed for resource-constrained environments. The approach focuses on reducing the computational demands of the LiteMedSAM model by adjusting the block depths of the image encoder, transformer depth, and dimensions in the mask decoder, and reducing the number of attention heads and the depth of the IoU head. However, the paper lacks empirical evidence to support the efficacy of the hyperparameter optimizations. There is a notable absence of comparative analysis with other state-of-the-art models in similar resource-constrained settings. The practical impact of the proposed optimizations on real-world clinical settings is not adequately demonstrated. While SwiftMedSAM proposes an approach to reducing the computational requirements of medical image segmentation models, the paper falls short in presenting a compelling case for its novelty and effectiveness.

---

### Official Review · Reviewer_KmWW · 2024-06-16
**This paper introduces an ultra-lightweight SAM model by optimizing the hyper-parameters of LiteMedSAM. Specifically, it reduces the block depth of the image encoder and decreases the transformer depth, MLP dimension, and attention head number in the mask decoder. The results show that the proposed SwiftMedSAM not only performs better overall but also significantly reduces parameter and inference time.**

**Rating:** 5
**Confidence:** 3

**Review:**

Firstly, this paper explores the tuning of structure hyper-parameters of LiteMedSAM in detail to improve the efficiency without lowering the segmentation performance compared to the baseline model. Besides, the model can also get a relatively good performance on some external datasets.

However, I think the novelty is limited. The main changes in the model are structural parameters, with no new pre-processing or post-processing techniques added. Some latest efficient image encoder can be explored further to increase the novelty.

And also I have two questions:

1. I have noticed discrepancies in the results for the baseline model, specifically in DSC, NSD, and efficiency, compared to the leaderboard and the original template. Could the author provide more details on how these results were obtained?

2. The value of model FLOPs seems excessively high. There might be an error in the calculation code. Could the author check and clarify that?

---

### Decision · Program_Chairs · 2024-10-01

Accept